# A 66–76 GHz Wide Dynamic Range GaAs Transceiver for Channel Emulator Application

**DOI:** 10.3390/mi13050809

**Published:** 2022-05-23

**Authors:** Peigen Zhou, Chen Wang, Jin Sun, Zhe Chen, Jixin Chen, Wei Hong

**Affiliations:** State Key Laboratory of Millimeter Waves, School of Information Science and Engineering, Southeast University, Nanjing 210096, China; chenwang@seu.edu.cn (C.W.); 220200844@seu.edu.cn (J.S.); zhechen@seu.edu.cn (Z.C.); jxchen@seu.edu.cn (J.C.); weihong@seu.edu.cn (W.H.)

**Keywords:** channel emulator, E-band, GaAs, ridged waveguide ladder transition, wide dynamic range

## Abstract

In this study, we developed a single-channel channel emulator module with an operating frequency covering 66–67 GHz, including a 66–76 GHz wide dynamic range monolithic integrated circuit designed based on 0.1 µm pHEMT GaAs process, a printed circuit board (PCB) power supply bias network, and low-loss ridge microstrip line to WR12 (60–90 GHz) waveguide transition structure. Benefiting from the on-chip multistage band-pass filter integrated at the local oscillator (LO) and radio frequency (RF) ends, the module’s spurious components at the RF port were greatly suppressed, making the module’s output power dynamic range over 50 dB. Due to the frequency-selective filter integrated in the LO chain, each clutter suppression in the LO chain exceeds 40 dBc. Up and down conversion loss of the module is better than 14 dB over the 66–67 GHz band, the measured IF input P1 dB is better than 10 dBm, and the module consumes 129 mA from a 5 V low dropout supply. A low-loss ridged waveguide ladder transition was designed (less than 0.4 dB) so that the output interface of the module is a WR12 waveguide interface, which is convenient for direct connection with an instrument with E-band (60–90 GHz) waveguide interface.

## 1. Introduction

6G is a new generation of mobile communication system developed for the needs of future mobile communication. According to the law of mobile communication development, 6G will have ultra-high spectrum utilization and energy efficiency, and will be an order of magnitude or higher more efficient than 5G in terms of transmission rate and spectrum utilization. Its wireless coverage performance, transmission delay, system security, and user experience will also be significantly improved [1]. Traditional mobile communications mainly use the frequency band below 6 GHz, which has become increasingly saturated. With the huge demand of 6G for system capacity, in some 6G applications, spectrum resources of several GHz may be required to meet specific transmission requirements. Due to the abundance of spectrum resources in the millimeter-wave (mm-wave) band, the World Radio Conference (WRC-19) has approved multiple mm-wave spectra for future mobile communications research and development, including the 66–67 GHz frequency range [2].

Different from frequency bands below 6G, the mm-wave band has poor penetration ability and large path loss. In order to effectively increase the propagation distance of the link, an effective solution is to adopt a large-scale multiple-input multiple-output (MIMO) system architecture [3]. In a MIMO system architecture, the number of antennas is greatly increased, and the systematic conductive testing becomes more impractical due to the influence of the long calibration time [4]. Moreover, the cost of testing is expensive, and the complexity of the test system increases exponentially. Therefore, multi-probe over-the-air (OTA) testing in millimeter-wave shielded anechoic chambers is becoming the mainstream testing solution [5,6,7].

As a vital part of the mm-wave OTA test system, a wireless channel emulator can accurately simulate and measure the wireless environment’s degradation of system performance, including the free space path loss, shadowing, and multi-path fading of the transmitted signals from the antenna ports [4]. Transceivers in the channel emulator send/receive up/down mm-wave signals to/from OTA probe antennas. The channel emulator is an important high-end general-purpose instrument for verifying the performance of system equipment and terminal equipment in various complex channel environments. However, due to high demands on dynamic range, compactness, cost, and power consumption, the mm-wave channel emulators are rarely seen, especially when operating at frequencies above 50 GHz. At present, the known channel emulator with the highest operating frequency is PROPSIM released by Keysight, which can support up to 43.5 GHz [8].

This paper presents a 66–67 GHz transceiver monolithic microwave integrated circuit (MMIC) in waveguide module for massive MIMO channel emulator application. The proposed transceiver is integrated by cascading a tripler chain for LO drive, a mixer, and a band-pass filter using a 0.1 µm pHEMT GaAs process.

A high dynamic output power range, up to 50 dB over 66-to-76 GHz, is achieved by carefully dealing with all unwanted harmonic signals employing highly selective band-pass and high-pass filters in the transceiver. Total power consumption of the chip is 645 mW with the supply voltage of 5 V. A low-loss ridged waveguide ladder transition was designed so that the output interface of the module is a WR12 waveguide interface, which is convenient for direct connection with an instrument with E-band waveguide interface. To the best of our knowledge, the proposed integrated module is a competitive E-band transceiver system for radio channel emulation application.

## 2. System Architecture

Figure 1 shows the system block diagram of the channel emulator and its connection to an RF system. The whole N-channel system includes three parts: baseband, LO signal source, IF chain, and RF front-end. Among them, the research on baseband (BB), Lo signal source, and IF TX/RX is relatively mature. However, there are few research reports on RF channel emulators. An E-band specific RF front-end chip for channel emulator applications is currently scarce in the market. Therefore, in this paper we develop a channel simulator for evaluating E-band channel characteristics for future E-band communication application scenarios.

The red dotted box in Figure 1 shows the system block diagram of the 66–67 GHz channel emulator module designed in this paper. The transceiver chip is realized by cascading a frequency tripler, the first LO band-pass filter (BPF), a LO driver amplifier, the LO high-pass filter (HPF), a mixer, and the second RF BPF [4]. The LO chain integrates a tripler instead of a doubler or higher order frequency multiplier in order to make a compromise between test convenience, conversion gain, and power consumption. Additionally, the tripler was chosen to have a trade-off between the cost of the LO signal source and power consumption. The IF frequency is selected around 27 GHz in the 5G mm-wave hotspot frequency band to facilitate the measurement of the channel emulator. In order to facilitate the connection with the instrument and the TX module, the IF and LO are coaxial interfaces, and the LO and IF ports of the transceiver are connected to the module via microstrip gold wire bonding. The RF output port of the module is a WR12 waveguide interface, as shown in Figure 1.

The mixer in the system architecture is a passive star mixer, so the channel emulator can be used for TX testing or RX testing. When applied to TX test scenarios, the RF signal output power budget is between −50 dBm and 0 dBm over 66-to-76 GHz. When an RX is tested, the input power range of the RF signal is −45~0 dBm. In addition, for the channel emulator, the transmit output power and receive noise figure are not key indicators, so the RF power amplifier and low noise amplifier are not integrated in the system [7,8]. To the best of our knowledge, extensive research has been carried out on mm-wave transceivers [9,10,11,12], while studies have rarely been published concerning E-band radio channel emulator application.

## 3. Circuit Design Methodology

In this section, we will present the circuit design methodology of the 66–67 GHz channel emulator, including the consideration of each block, and then present the simulated and measured results of key building blocks.

### 3.1. Frequency Tripler

As shown in Figure 1, the LO chain integrates a frequency tripler, a BPF, an LO driving power amplifier, and an HPF. The schematic of the frequency tripler and the succeeding frequency selective BPF is illustrated in Figure 2. The tripler core is composed of anti-parallel diode pairs (APDP) [13,14], and the diode is implemented by connecting the drain and source of pHEMT transistors as the cathode with the gate as the anode. Due to the passive structure, sufficient input power is required for the tripler to generate odd-order harmonics while suppressing even-order spurs [15]. Two 4-finger 10 µm pHEMTs are employed as the APDP in this design in consideration of a trade-off between output power and bandwidth. The input matching network consists of a capacitor connected in parallel to ground and a microstrip line connected in series. The output matching network consists of two capacitors connected in series and an inductor connected in parallel to ground in a T-shaped configuration. In order to effectively improve the unwanted harmonics suppression of the tripler and reduce the frequency conversion loss, a transmission line TL as depicted in Figure 2 is adopted to reflect idle frequency signals to the APDP core. A seventh-order BPF was connected after the frequency multiplier to further improve the harmonic suppression characteristics of the LO chain.

The simulation results of each harmonic output power of the tripler plus the cascaded BPF when the input power is 16 dBm are shown in Figure 3. The simulated input and output return loss are better than −10 dB over 41–51 GHz, and the simulated output power of the third harmonic signal is around 0 dBm. Benefiting from the BPF, all unwanted harmonics can be suppressed significantly within the interested frequency bands. It can be deduced from Figure 3 that the fundamental signal and 2nd harmonic rejection are over 30 dB compared with the third harmonic signal, and the 4th and 5th harmonic suppressions are better than 35 dB.

### 3.2. Power Amplifier

The schematic of the LO driving power amplifier and the HPF is shown in Figure 4. The power amplifier adopts a three-stage common-source structure to obtain sufficient power gain at 41–51 GHz while ensuring high power-added efficiency (PAE) [16]. The first two driver stages use a 4 × 25 µm pHEMT transistor to obtain sufficient gain, and the final power stage uses a larger 4 × 50 µm pHEMT transistor to obtain a sufficiently high output power. Source degeneration inductors are connected to the source of the pHEMT transistors to increase the stability of the power amplifier. The input matching network of the power amplifier is co-designed with the previous frequency selective BPF. The output matching network is co-simulated with the following HPF in full wave electromagnetic simulation. All transistors are biased with a shunt by-pass capacitor and a series resistor close to the gate, and the dc power (Vdd) is feeded through an LC network to the drain.

For the power amplifier used in the LO chain, a key indicator is the out-of-band suppression. For the channel emulator, the requirements for the suppression of each harmonic in the LO chain are higher, because this out-of-band clutter will degrade the dynamic operating power range of the channel emulator. Therefore, the fifth-order HPF is employed after the power amplifier. As depicted in Figure 5, an additional 25 dBc forward fundamental signal rejection can be obtained with the HPF. The 2nd harmonic is suppressed by 10–25 dBc, while the 4th and 5th harmonics are amplified with low gain by tuning matching networks to make the out-of-band gain drop slope as steep as possible. The simulated input and output return loss of the power amplifier is better than −10 dB in the 41–51 GHz frequency band, and the small-signal gain is around 20 dB. Saturated output power varies from 17 dBm to 18.8 dBm including the insertion loss of the HPF. The circuit draws a total current of 102 mA at 3.3 V power supply with −0.4 V gate bias.

### 3.3. Star Mixer

The passive star mixer has been widely used in mm-wave on-chip systems since being proposed by Basu and Maas [17]. Its highly symmetrical star topology can offer better port-to-port isolation and high spurious rejection [18,19]. Different from the traditional star mixer, the traditional straight symmetrical Marchand Balun is modified to S type with two bends, which can reduce the chip size while ensuring the performance of the mixer, as shown in Figure 6. To reduce the coupling between transmission lines, the decoupling ground wall consisting of metal-vias array is inserted between four S type balun and the IF port. Additionally, double side coupling lines of the Marchand Balun are grounded separately in the end, rather than joint together by the bottom metal layer for better spurious rejection. Four diodes measuring 1 × 15 µm are arranged in a symmetrical layout for broad IF bandwidth. In addition, the LO input matching network is co-optimized with a pre-stage HPF output network. Furthermore, an extra BPF with the same topology as the one after the tripler is integrated after the star mixer. Finally, the micrograph of the complete channel emulator is shown in Figure 7, with a chip size of 2.7 by 0.9 mm^2^.

Simulation results of the mixer with the following BPF show that both the input and output return loss is better than −10 dB, and the conversion loss of the mixer is about 6.5 dB from 66 to 76 GHz. Furthermore, the insertion loss of the BPF after the mixer is between −1.4 and −2 dB within the RF operating bandwidth.

## 4. Packaging Methodology

As an important part of the OTA test system, the channel emulator module is directly connected to the RF transceiver of the communication system to simulate path loss, multipath fading, and shadow fading in the wireless environment. Therefore, the input and output interface of the channel emulator should have better robustness so that it can be directly connected with the transceiver or instrument. In mm-wave low-frequency bands (below 30 GHz), interfaces such as instrument or RF transceiver outputs typically use coaxial connectors [20]. When the frequency is higher than 50 GHz, the instrument interface is usually designed as a waveguide interface for more favorable stability and cost.

This channel emulator is aimed at 66–67 GHz wireless communication system test applications, and the RF frequency range is 66–67 GHz. The IF port is compatible with the existing commercial 5G mm-wave band, the IF frequency is 27 GHz, and the IF power coverage range is −40~10 dBm. The LO input frequency range is 13~16.33 GHz. Therefore, the LO and the IF port use a coaxial connection scheme, and the RF port uses a WR12 waveguide interface. The frequency of the IF and LO ports is lower, and the influence of the gold bonding wire is small. In the module design, the RF and IF signal interfaces on the chip are directly bonded to the PCB through gold wires, and are connected to the coaxial connector through the 50 ohm characteristic impedance microstrip line on the PCB.

The working frequency of the RF port is relatively high. In order to transfer the RF port to the WR12 waveguide interface, firstly, the RF GSG PAD-to-50ohm microstrip line (as shown in Figure 8) was designed on the TLY-5 sheet with a dielectric constant of 2.2 and a thickness of 0.254 mm using bonding wires. The distance from the microstrip line to the edge of the RF GSG PAD, the height of the gold bonding wire, and the structure of the microstrip line were all optimized by electromagnetic simulation [21]. Then, a low-loss microstrip to WR12 ridged waveguide ladder transition (RWLT) structure (as shown in Figure 9) was designed using aluminum metal [22,23]. The transition structure includes a 4-stage stepped impedance transformation. By optimizing the length and height of each stepped transformation unit, a lower transition loss from the microstrip to WR12 can be obtained, and the connection loss in the 66–67 GHz frequency band is lower than 0.36 dB.

Figure 10 shows the S-parameters of the transition structure between the simulated RF GSG PAD and the microstrip line and the microstrip line to the WR12 waveguide port in the entire E-band. As illustrated in Figure 10, in the channel emulator operating frequency band 66–67 GHz, the return losses (S11, S22) of the above connection structures are all better than −10 dB, and the total insertion loss between the RF GSG PAD and the WR12 waveguide port is equal to less than 2 dB.

The photograph of the opened split-block and assembled module including the RF transceiver chip, RF microstrip-to-waveguide transition, and the DC biasing network is shown in Figure 11. The PCB was sintered on the aluminum structure to obtain a tighter and better grounding effect. The RF transceiver chip was installed in a groove dug in the middle of the PCB. The depth of the groove was slightly higher than the height of the chip to ensure that the top surface of the chip (after the conductive adhesive is pasted) was the same height as the PCB surface. In the design of the DC bias network, we first placed some decoupling chip capacitors next to the chip to obtain better DC bias characteristics. The DC voltage of the amplifier’s drain, gate, etc. is provided through the LDO DC biasing network, and the entire module has only a 5V DC input voltage. On the side of the module, two 2.92-mm coaxial connectors were used as LO and IF ports, and the RF port is a WR12 waveguide interface.

## 5. Measurement Results

The measurements were performed using two steps. First, the RF transceiver chip was measured with on-chip characterization to obtain accurate conversion gain, dynamic range, and other performances; then, the packaged module was tested. In addition, since the harmonic suppression performance of the LO link is important for the channel emulator, the LO chain was separately processed and the output power of each harmonic was tested; the measured results are illustrated in Figure 12. In the working frequency range of 3×LO_In_ (39–49 GHz), the output power of the third harmonic of the LO chain exceeds 15 dBm, and the unwanted harmonics suppression of each order exceeds 40 dBc.

The on-chip measurement setup of an up-conversion configuration is shown in Figure 13. In this measurement, the chip is implemented as a transmitter. LO and IF signals are pumped from two signal source Keysight E8257Ds through co-axial cables and probes. The RF signal is measured by a waveguide probe and a spectrum analyzer. As our corresponding band was E-band, the measurements were conducted by two steps and limited by our testing equipment. One is the V-band (50–75 GHz) test and the other is the W-band (75–110 GHz) measurement. For the two setups, different waveguide probes and two kinds of harmonic mixers were applied. To obtain the accurate output power of the chip, power meter was employed. The down-conversion test setup was similar to that in Figure 13, expect that the RF output was changed to input, and the IF input was changed to output.

Figure 14 shows the measured results of the conversion gain of the RF transceiver chip and the channel emulator module when the LO signal power is 16 dBm and the IF signal frequency is 27 GHz. The measured results show that the conversion gain of the RF chip is between −8 and −12 dB in the 66–67 GHz frequency band. The conversion gain corresponding to the transmit mode and the receive mode are in agreement. The conversion loss measured in the 66–67 GHz band is large, mainly because the output power of the LO chain is low, which is not enough to drive the star mixer. The conversion loss of the channel emulator module integrated with the RF waveguide interface and the LO and IF coaxial interfaces is about 2 dB higher than the on-chip measured results. In addition to the transition between the RF GSG PAD and the WR12 waveguide port, this part of the loss also includes the loss of a section of the IF signal transmission line on the PCB.

Additionally, the in-band harmonic signal has a great influence on the sensitivity of the channel emulator. In this design, the frequency of the IF signal is 27 GHz, the frequency of the LO signal is 13–16.33 GHz, and the frequency of the RF signal is 66–67 GHz. The 5th harmonic of the LO signal falls within the RF band. Therefore, we measured the power of the 5th harmonic of the LO input signal in the RF band, and the results are shown in Figure 14. It can be deduced that within the RF bandwidth, the 5th harmonic leakage amplitude of the LO is less than −63 dBm.

Figure 15 shows the measured results of the RF output power of the channel simulator module varying with the IF input power. The measured input P1 dB of the module is about 10 dBm. In the 66–67 GHz band, the RF output 1 dB compression point is between −2 and −4.5 dBm. For linear channel simulator transceiver application scenarios, considering a certain spurious suppression margin, the dynamic operating power range of the module can reach more than 50 dB. In the down-conversion test, the receiving dynamic range can reach more than 54 dB under the same signal quality.

## 6. Conclusions

This paper presents a 66–67 GHz channel emulator module in a waveguide package. The module includes a 66–67 GHz transceiver chip processed by a 0.1 μm pHEMT GaAs process, a DC bias network designed on a PCB, and a low-loss ridge microstrip line to a WR12 waveguide transition structure. In each circuit block of the transceiver, the various clutter signals are closely monitored and studied. By integrating multiple frequency-selective filters and a high-isolation mixer in the link, the transceiver achieves good spurious rejection performance. In addition, benefiting from the designed low-loss stepped microstrip-to-waveguide transition structure, the RF output of the module is a WR12 waveguide interface. This makes it easy to interface with commercial E-band instruments to evaluate the performance of the channel emulator module. Measured results show that the channel emulator module can achieve a dynamic operating power range of more than 50 dB in the 66 to 76 GHz frequency band. Due to better dynamic range performance and higher integration, the transceiver chip and module can meet the application requirements of E-band channel simulators.

## Figures and Tables

**Figure 1 micromachines-13-00809-f001:**
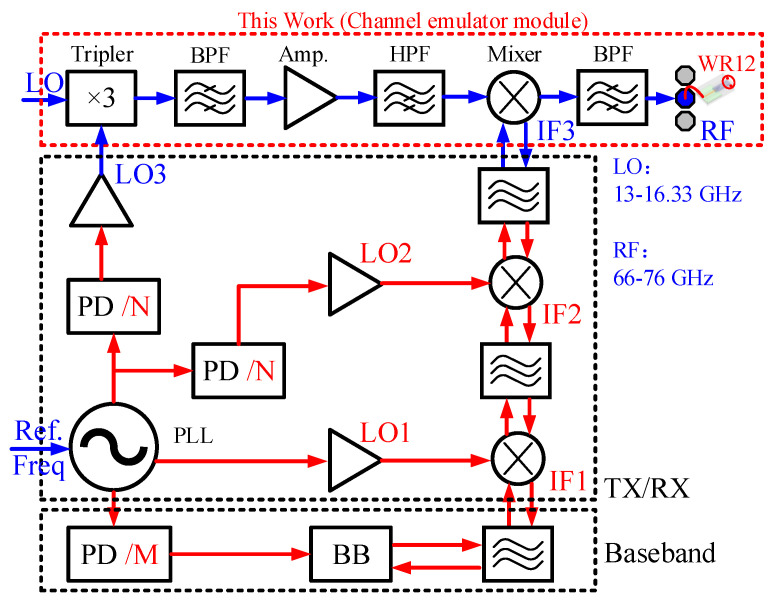
System block diagram of the channel emulator system.

**Figure 2 micromachines-13-00809-f002:**
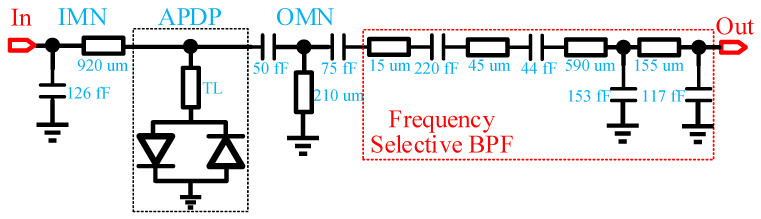
Schematic of the frequency tripler and its cascaded frequency selective BPF.

**Figure 3 micromachines-13-00809-f003:**
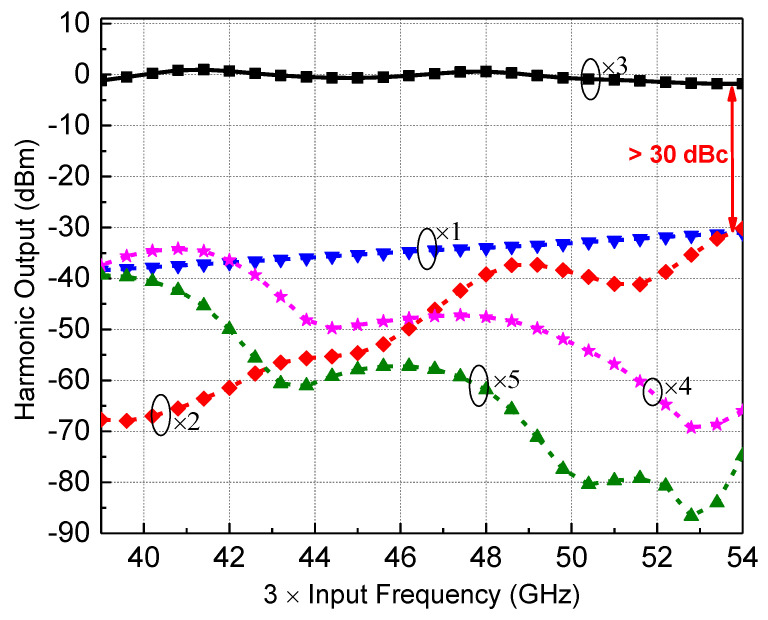
Simulated output power of the tripler with BPF.

**Figure 4 micromachines-13-00809-f004:**
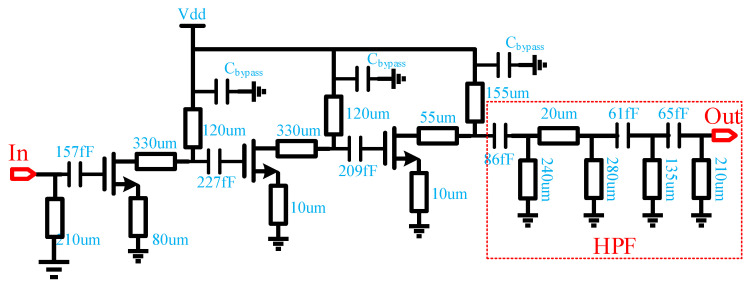
Schematic of the power amplifier with HPF.

**Figure 5 micromachines-13-00809-f005:**
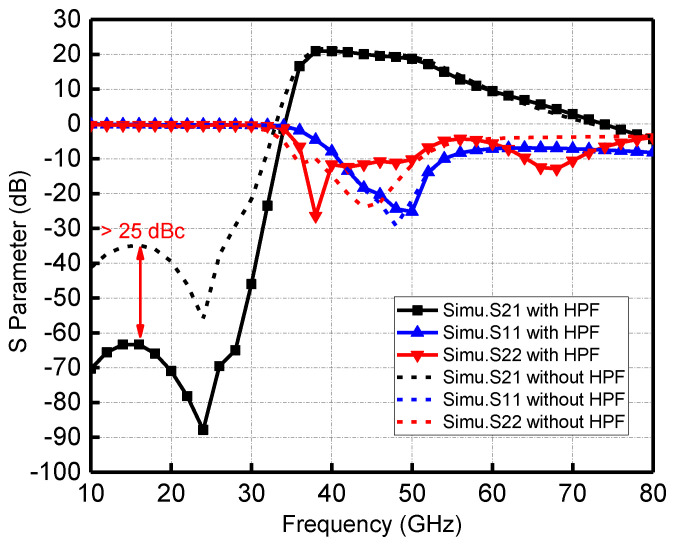
Simulated S-parameters of the LO driver power amplifier with or without HPF.

**Figure 6 micromachines-13-00809-f006:**
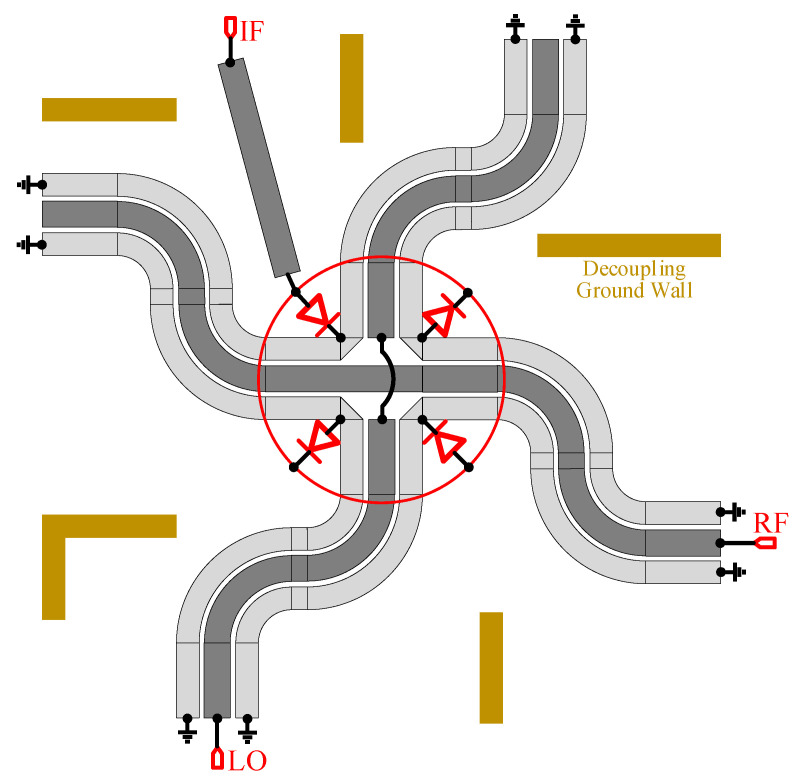
Schematic of the modified star mixer.

**Figure 7 micromachines-13-00809-f007:**
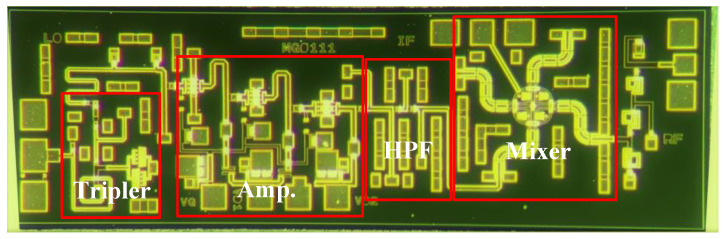
Chip micrograph of the channel emulator.

**Figure 8 micromachines-13-00809-f008:**
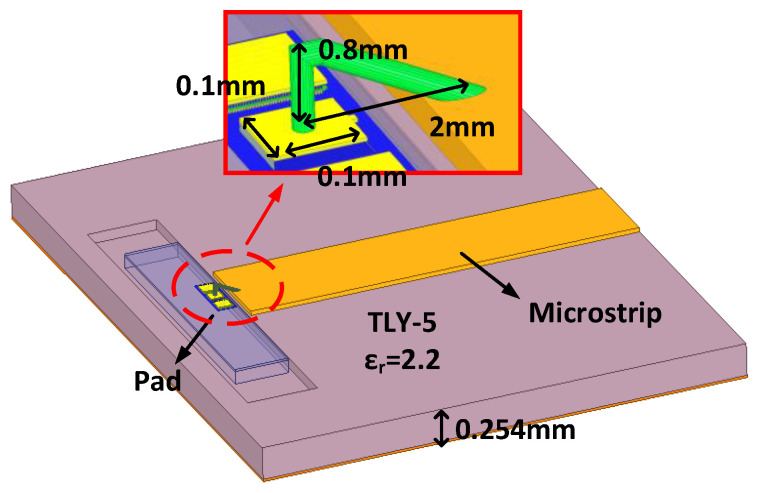
3D model of the RF GSG PAD to microstrip transition.

**Figure 9 micromachines-13-00809-f009:**
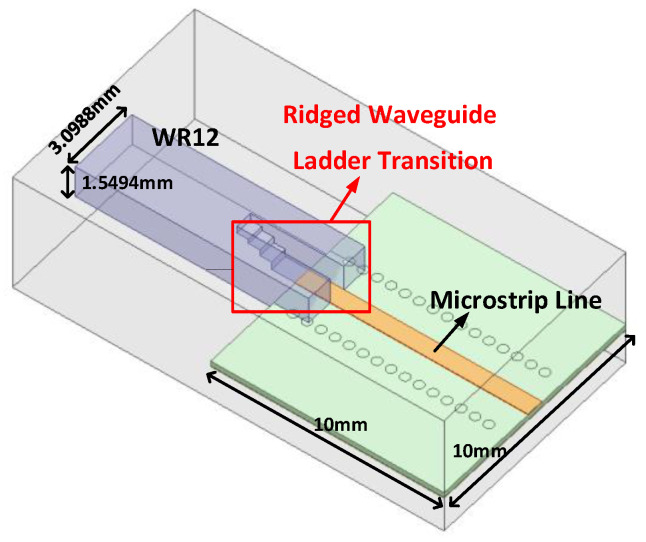
Ridge waveguide ladder transition from WR12 waveguide interface to 50-ohm microstrip line.

**Figure 10 micromachines-13-00809-f010:**
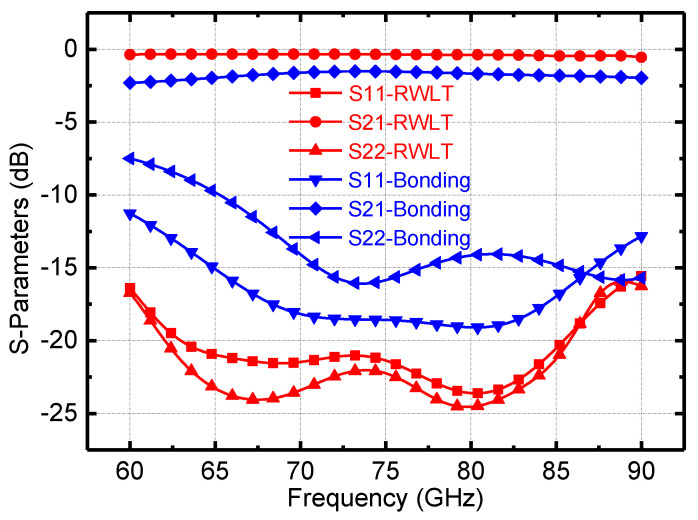
The simulated connection loss of the RF GSG PAD to the microstrip line and the microstrip line to the WR12 waveguide port.

**Figure 11 micromachines-13-00809-f011:**
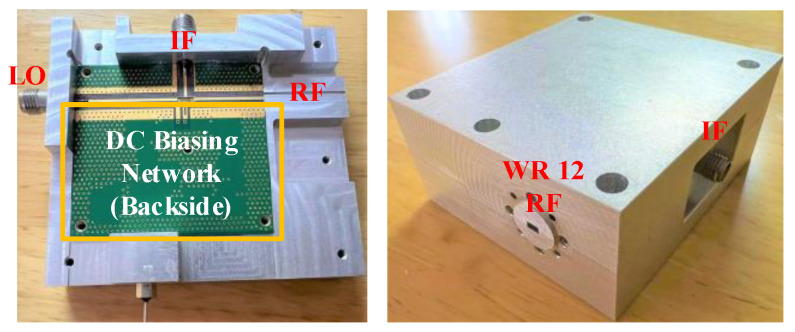
Photo of the 66–67 GHz channel emulator module integrating the RF transceiver chip, the microstrip-to-waveguide transition structure, and the DC biasing network.

**Figure 12 micromachines-13-00809-f012:**
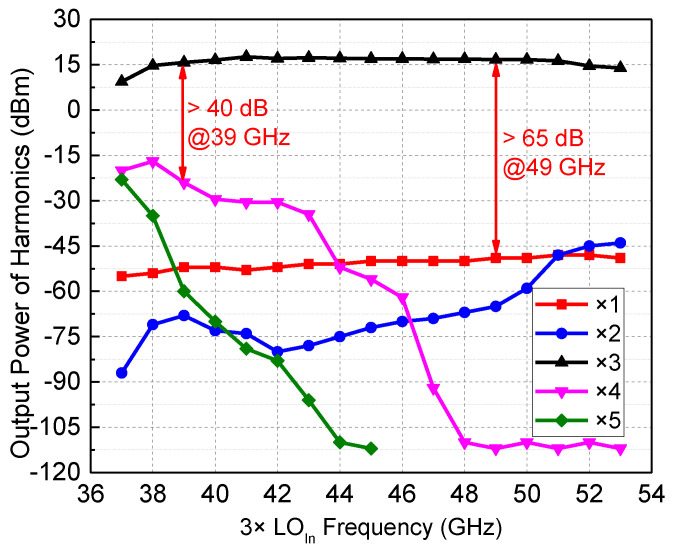
The measured output power of each harmonic of the LO chain.

**Figure 13 micromachines-13-00809-f013:**
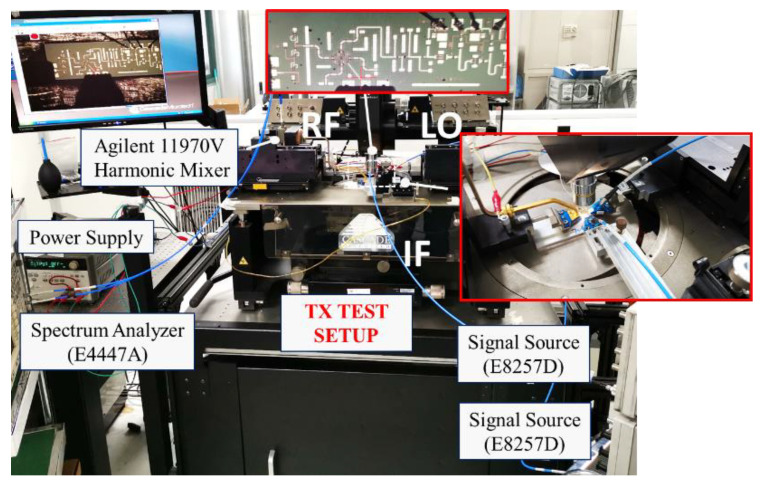
On-chip measurement setup.

**Figure 14 micromachines-13-00809-f014:**
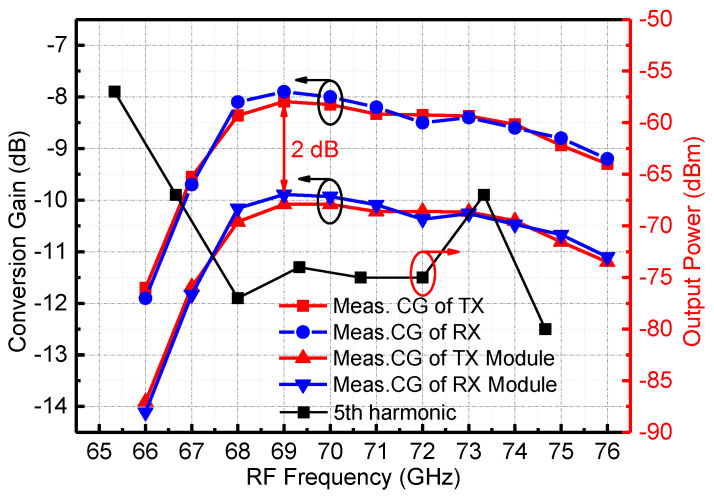
Measured conversion gain of the RF channel emulator and the module.

**Figure 15 micromachines-13-00809-f015:**
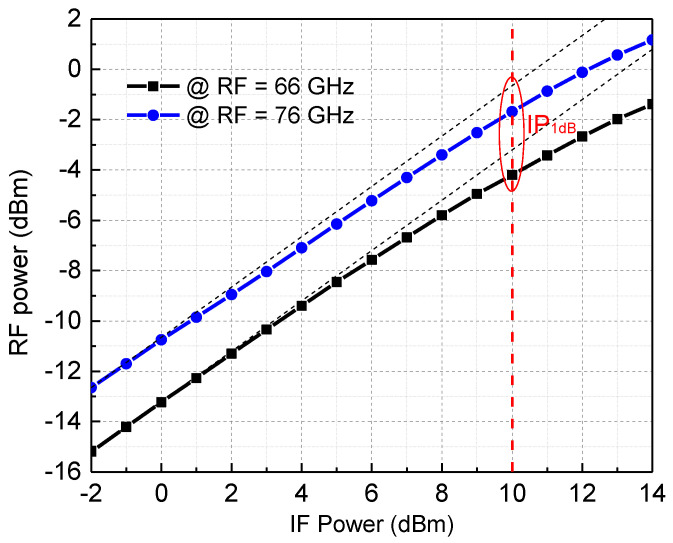
The measured RF output power of the channel emulator varies with the IF input power.

## Data Availability

Not applicable.

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
