# Peer review of "A 66–76 GHz Wide Dynamic Range GaAs Transceiver for Channel Emulator Application"

_micromachines, 2022, doi:10.3390/mi13050809_

Round 1
Reviewer 1 Report
The paper describes a channel emulator module operating in the frequency band of 66-76 GHz. The authors provide detailed information and characterization of the several blocks validating their conclusions with results. Although each element of the emulator may not be considered novel, except perhaps the star mixer, the integration of the system is challenging, deserves relevance and is of interest to the research community. The reviewer believes that the paper would benefit from a description of how the proposed architecture can in fact emulate the wireless channel and the relevant parameters to that purpose. Also, a conclusion at the end of the paper is missing.
Author Response
The authors would like to express our sincere appreciation to the Assistant Editor and reviewers for their time and efforts in providing valuable comments and suggestions. The reviewers’ comments had been carefully studied and we have revised our manuscript to address all the concerns being raised. The Response to Reviewers' comments are shown in the attachments.

Reviewer 2 Report
1. The manuscript has ignored the instructions for authors. The sections are needed to create according to the guideline in "Research Manuscript Sections".
2. Abstract needs to modify be more quantitative. You can absorb readers' consideration by having some numerical results in this section. Moreover, numerical results should be presented in the conclusion.
3. The authors need to paragraph the reminder section at the end of the introduction.
4. It helps to appreciate the paper by having a related work section. The authors should consider more recent research done in the field of their study (especially in the years 2020 and 2021 onwards).
5. The authors should clearly describe related review in more detail, contrasting the limitations of the related works. If possible, the authors can give a table pinpointing the advantage or limitations of each work.
6. There is no discussion of user requirements, technological options and support for the decisions made at the design. The authors should include more technical details and explanations.
7. Numerical results in this paper are not enough to support the conclusions. The comparison to other improved schemes (within the last 3 years) is required.
8. The authors did not provide solid achievements in this manuscript since this paper seems to be a somewhat incremental piece of work based on earlier research results [4].
Author Response
The authors would like to express our sincere appreciation to the Assistant Editor and reviewers for their time and efforts in providing valuable comments and suggestions. The reviewers’ comments had been carefully studied and we have revised our manuscript to address all the concerns being raised. The response are shown in the attachments.

Round 2
Reviewer 2 Report
This paper has edited and revised according to the reviewer's suggestions.